# Emerging Marine Biotoxins in European Waters: Potential Risks and Analytical Challenges

**DOI:** 10.3390/md20030199

**Published:** 2022-03-08

**Authors:** Paz Otero, Marisa Silva

**Affiliations:** 1Department of Pharmacology, Pharmacy and Pharmaceutical Technology, Faculty of Veterinary Science, Universidade de Santiago de Compostela, 27002 Lugo, Spain; 2MARE—Marine and Environmental Sciences Centre, Faculty of Sciences, University of Lisbon, Campo Grande, 1749-016 Lisbon, Portugal; 3Department of Plant Biology, Faculty of Sciences, University of Lisbon, Campo Grande, 1749-016 Lisbon, Portugal

**Keywords:** emerging toxin, European waters, poisoning risks, detection methods

## Abstract

Harmful algal blooms pose a challenge regarding food safety due to their erratic nature and forming circumstances which are yet to be disclosed. The best strategy to protect human consumers is through legislation and monitoring strategies. Global warming and anthropological intervention aided the migration and establishment of emerging toxin producers into Europe’s temperate waters, creating a new threat to human public health. The lack of information, standards, and reference materials delay effective solutions, being a matter of urgent resolution. In this work, the recent findings of the presence of emerging azaspiracids, spirolildes, pinnatoxins, gymnodimines, palitoxins, ciguatoxins, brevetoxins, and tetrodotoxins on European Coasts are addressed. The information concerning emerging toxins such as new matrices, locations, and toxicity assays is paramount to set the risk assessment guidelines, regulatory levels, and analytical methodology that would protect the consumers.

## 1. Introduction

Marine biotoxins are natural toxic metabolites usually produced during harmful algal blooms (HABs) that accumulate in marine organisms and migrate along the food chain [1]. A HAB is characterized by a rapid proliferation of phytoplankton, the so-called red tides. As yet, their forming circumstances have not been disclosed, although the rise of water temperature and anthropological intervention are pointed to as main triggers [2]. These secondary compounds are produced to give their producers a competitive advantage against similar species and also to provide a defense against predators [3]. Their deleterious effects can impact an entire ecosystem, leading to high fish mortality, affecting fisheries and aquaculture, and threatening public health [4]. The main entrance route to humans of these toxic substances is through consumption of contaminated seafood and a high number of intoxications occur every year [5,6]. The occurrence of these marine biotoxins can cause massive economic losses to the fishery and aquaculture industry due to the cautionary closure of fishing and cultivation areas [7]. Hence, their harmful effects and socio-economic consequences have prompted the elaboration and establishment of strong European legislation and monitoring protocols to detect and characterize them and fix their maximum levels in seafood [8,9]. However, this has not occurred for all known marine toxins, and some groups still need to be regulated. In Europe, the legislated group of lipophilic marine toxins consists of four different chemical groups: yessotoxins (YTXs), azaspiracids (AZAs), pectenotoxins (PTXs), and okadaic acid (OA) and respective derivatives. To date, a total of 13 analogues are legislated (OA, DTX-1, DTX-2, DTX-3, PTX-1, PTX-2, AZA-,2, AZA-2, AZA-2, YTX, 45-OH-YTX, homoYTX, 45-homoYTX) [8]. However, PTX has been removed from the health standards for live bivalve molluscs in Commission Delegated Regulation (EU) 2021/1374 [10], since the European Food Safety Authority (EFSA) has assessed their risk and stated that there is no evidence of adverse effects in humans linked with this toxin [11]. Therefore, it would be convenient to remove them from the Implementing Regulation (EU) 2019/627 [12] (Figure 1). The legislated group of hydrophilic toxins is comprised of two distinct groups: saxitoxin (STX) and domoic acid (DA) and their derivatives. The STX group is composed of four subgroups: the C group, N-sulfocarbamoyl-gonyautoxins 1–4, (C1, C2, C3 and C4); decarbamoyl gonyautoxins 1–4 (dcGTX1, dcGTX2, dcGTX3 and dcGTX4); GTXs group, gonyautoxins 1–5 (GTX1, GTX2, GTX3, GTX4, B1, and B6), and the STX group (STX, decarbamoyl saxitoxin (dcSTX), and Neosaxitoxin (NEO)) [13,14] Regarding the DA group, there are ten identified DA isomers: isodomoic acids A, B, C, D, E, F, G, and H (iso-DA (A–H), and the isomers DA C5′-diastereomer and *epi*-DA [15]. For these toxins, maximum levels in shellfish were determined for human consumption using technology recognized as the state-of-the-art reference for the detection of these marine toxins using chemical methods [16]. Until 2013, for the legislated toxin group of lipophilic toxins, all official toxicity determinations in EU countries were carried out through mouse bioassay (MBA), but due to ethical and technical concerns, this method progressively fell into disuse and toxin determinations were replaced by chromatographic hyphenated techniques, supported by the update of the legislation [14,15,17]. Thus, the use of analytical techniques is encouraged to be applied as the reference method, following indications agreed upon by the National Reference Laboratories Network [18].

Therefore, the official monitoring systems were elaborated to ensure the safety levels of legislated compounds in seafood for human consumption, determining the official methodologies for their detection and quantification [19]. In these regulations, information can be found on which toxins to monitor and their legal limits. However, there is a rising new concern: emerging marine biotoxins are appearing in new areas, aided by the effects of globalization and climate change [7], which facilitates the migration and establishment of non-indigenous toxin producers [20]. Among these emerging marine biotoxins, cyclic imines (CIs), which comprise the lipophilic compounds spirolides (SPXs), gymnodimines (GYMs), and pinnatoxins (PnTXs), may entail harmful effects on human health. Another example is the AZAs group which nowadays consists of more than 60 compounds with different toxicity [21]. In addition, the presence in Europe of toxins from other latitudes like ciguatoxins (CTXs) increases the need for continued renovation of the information available and a methodology update. Having the same molecular target, brevetoxins (BTXs), responsible for neurotoxic shellfish poisoning syndrome, have become more prevalent in European waters in recent years [21]. Tetrodotoxins (TTXs) were first reported in Europe in 2008 [22], and, since then, the number of reports has increased substantially [23,24,25,26]. Even though human poisoning events have not been associated with some groups, their toxicity has already been proven in animal models or cell assays [27,28]. Additionally, the effects of chronicle exposure are scarce or absent, with unknown consequences [28,29,30,31,32,33]. The scarcity of reports could be also be due to their novelty in European territory, as health professionals are not aware of this new threat [34].

To ensure consumers’ safety from these emerging toxins, efforts are being made regarding the development of new detection methodologies for the update of toxin monitoring techniques [20,35]. The preferred methodology is based on in vitro or chemical approaches since biological assays using mice fell into disuse due to lack of accuracy and ethical issues [36,37]. Yet, the analysis by LC-MS/MS proves to be a very challenging task [38], due to the scarcity of reference materials, narrowing the number of targeted compounds, and underestimating public health risk. In this review, the current situation of the so-called “emerging toxins” in the EU is addressed, including the discovery of compounds in new areas and matrices, their frequency, and the analytical challenges for their detection.

## 2. Emerging Marine Toxins in European Waters and Their Risks

### 2.1. Imine Cyclic Toxins

#### 2.1.1. Spirolides

Spirolides (SPXs) are macrocyclic compounds with imine and spiro-linked ether moieties (Figure 2C) [39]. Together with pinnatoxins and gymnodimines, they belong to the family of CI, an emerging group of lipophilic marine toxins. In particular, SPXs are a major cause of concern, due to their global distribution. They are produced by the dinoflagellates *Alexandrium ostenfeldii* or *Alexandrium peruviaunum* [40]. SPXs were firstly identified in extracts from the digestive glands of mussels (*Mytilus edulis*) and scallops (*Placopecten magellanicus*) on the Atlantic coast of Nova Scotia (Canada), in the early 1990s, being later detected in Europe, in 2005 [41]. Since then, the presence of CIs has gradually increased, with 13-desmethyl spirolide C (SPX-13) being the most extended analogue [42]. SPXs are a highly heterogeneous group with compounds that vary among strains from different locations [40]. In total, 16 SPX analogues have been detected in European and South and North American waters, including SPX-13, 13,19-didesmethyl spirolide C (SPX-13,19), 20-methyl spirolide C (SPX-20), Spirolide A-I, 27-hydroxy-13,19-didesmethyl SPX-C, 27-hydroxy-13-desmethyl SPX-C and 27-oxo-13,19-didesmethyl SPX-C [43,44,45]. All these toxins are proven to accumulate in shellfish through microalgae feeding. Their presence is common in edible species like mussels, clams, and cockles. For example, SPX-13 concentrations in commercial Galician mussels (*Mytillus galloprovinciales*) were reported up to 28.9 µg/kg [6,46]. Furthermore, in Europe to date, SPXs have also been found in gastropods such as *Gibbula umbilicalis*, *Nucella lapillus*, *Patella intermedia*, *Monodonta* sp., and starfish like *Marthasterias glacialis* [6,47] as well as in food supplements based on mussel extracts of *Perna canaliculus* from New Zealand [48]. Recently, nine new SPX with a structure of triketal rings have been proposed, one of them structurally close to PTXs [49]. However, due to the unavailability of high amounts of these toxins with high purity, neither identification by NMR nor toxicity studies were performed.

Despite their frequent occurrence in shellfish, no human intoxication has been reported linked to SPXs consumption [39,42,50]. However, their fast-acting toxicity following intraperitoneal injection in mice has led to concern over their human health implications [44]. Their mode of action is based on the interaction with muscle-type and neuronal nicotinic acetylcholine (ACh) receptors (nAChR) [50]. For this reason, the European Union Reference Laboratory (EURL) working group on toxicology proposed a guidance level of 400 µg SPXs/kg of shellfish [29] and an oral LD_50_ for SPX-13 of 130 μg/kg, and an i.p. LD_50_ of 7–28 μg/kg was established [39].

#### 2.1.2. Pinnatoxins

The group of pinnatoxins (PnTXs) consists of eight analogues, pinnatoxin A-H, (PnTX A-H), whose chemical structure is very similar to that of SPXs [29]. The only producer organism of these toxins described so far is *Vulcanodinium rugossum* (*V. rugossum*). PnTXs were detected for the first time in Japanese molluscs *Pinna muricata* in 1995, with PnTX-A being the identified molecule [51]. After the first episode in Japan, they were reported in further locations and different species like Pacific oysters (*Crassostrea gigas*) and razorfish (*Pinna bicolor*) from South Australia and Northland, New Zealand [52,53]. Currently, these compounds are being detected in oysters, mussels, razor clams, and clams from coasts of numerous countries such as Canada, [54], Norway [55], France [56], Spain [6,20,57], and Chile [48], proving their wide distribution. It seems that dinoflagellates from the Pacific Coast of New Zealand, Australia, and the Atlantic coast of Cuba mainly produce PnTX-E and PnTX-F [52,53,58]. The microalgae from European waters mainly exhibits a different toxin profile, consisting in the production of PnTX-G and PnTX-A [6,46,57,59]. Concentrations of PnTXs in molluscs are generally low, around 10–40 µg/kg [6,48]; nevertheless, PnTX-G in French mussels reached 1.2 mg/kg in an episode in 2010 [44,60]. In fact, due to the recurrence of these toxins in the French coast, the French Agency for Food, Environmental and Occupational Health and Safety fixed an acceptable contamination level of 23 μg PnTX-G/kg [61], and in vivo toxicity assays have set an oral LD_50_ for PnTx-G of 208 μg/kg and a provisional no-observed-effect level (NOEL) of 120 μg/kg [62]. Like SPXs, no human intoxications have been reported linked to PnTXs consumption. However, last year sixty people needed medical assistance due to a dermatitis outbreak in Cienfuegos (Cuba) after direct exposure to seawater containing a bloom of *V. rugossum* [58]. Patients were treated with antibiotics and some children were hospitalized. After that, it was confirmed that *V. rugosum* cells contained mainly PnTX-F (441.8 fg/cell), PnTX-E (94.2 fg/cell), and small concentrations of PnTX-D, PnTX-G, and some isomers of PnTX-E and -F [58].

#### 2.1.3. Gymnodimines

Gymnodimines (GYMs) were first detected in oysters of the species *Tiostrea chilensis* in New Zealand in the early 1990s [63]. The first analogue isolated was GYM-A produced by the dinoflagellate *Karenia selliformis* (*K. selliformis*). After that episode, two different compounds (GYM-B and GYM-C) were identified in New Zealand cells from the same organism [64,65]. Today, eight GYM analogues have been identified [66]. It was recently confirmed that GYMs shares with SPX the same producer, the dinoflagellate species of *Alexandrium* [59,67]. The fact suggests the existence of common biosynthesis pathways for the production of these biotoxins, between the species *Karenia selliformis* and *Alexandrium ostenfeldii* [63]. GYMs have been detected in different matrices like mussels (*M. galloprovincialis*), oysters (*T. chilensis*), scallops (*Pecten novaezelandiae*), and clams (*Ruditapes decussatus*) worldwide [63,67,68]. In Europe, GYM-A was recently found for the first time in Italian mussels, in 84% of the samples, reaching the concentration of 12.1 µg kg^−1^ [66]. In addition, GYM A was also found for the first time in several molluscs from the north Atlantic Coast of Spain, including mussels (*Mytilusgalloprovincialis*), cockle (*Cerastoderma edule*), and oysters (*Magallanagigas*, and *Ostrea edulis*) at the maximum level of 23.93 µg/kg [69].

### 2.2. Azaspiracids

Azaspiracids (AZAs) are lipophilic molecules with a structure consisting of a cyclic amine, three spiro-type ring bonds, and a carboxylic acid group (Figure 2A) [70]. They were first identified after a poisoning incident, in the Netherlands, where at least eight people became sick after ingesting mussels from the species *Mytilus edulis* collected in Killary Harbor, on the western coast of Ireland [71]. The responsible analogue was identified as AZA-1 and the main symptoms were nausea, vomiting, diarrhea, and stomach cramps [72]. Today, the AZAs group consist of more than 60 compounds [21] and is mainly produced by dinoflagellates from genera *Azadinium* and *Amphidoma* [73]. For example, the species *Azadinium poporum* was proofed to produce AZA-2, -11, -36, -37, -40, -41, -42, -59, and -62, while *Azadinium spinosum* releases AZA-1, -2, -11, 33, -34, -35, -50, and -51; *Azadinium dexteroporum* produces *epi*-AZA-7, AZA-35, 54, -55, -56, -57, and -58; and *Amphidoma languida* produces AZA-2, -38, -39, -43, -52, and -53 [70]. Some other AZAs are metabolites and products from oxidation, hydroxylation, decarboxylation, and dehydration occurring in shellfish [21,74]. AZA-17 and AZA-19 were found to be the main mussel metabolites of AZA-1 and AZA-2, respectively. AZA-3, AZA-4, AZA-6, and AZA-9 are formed via heat accelerated decarboxylation of AZA-17, AZA-21, AZA-19, and AZA-23, respectively [75]. The mode of action of AZAs in humans is unknown, although in vivo studies in mice showed that AZA1 is absorbed and distributed, being detected in spleen, kidney, lung, heart and liver, and brain [5,76]. AZA1 has been found to have cardiotoxic potential in rats. After repeated i.p. administration of sublethal doses, the rats displayed signs of heart failure and alteration of myocardium structure [77]. Oral toxicity of AZA1 towards mice indicates that single oral doses causing lethality vary from 250 to 600 μg/kg Some studies showed the oral toxicity of AZA1 [78,79,80]. Oral toxicity of AZA1 towards mice indicates that single oral doses causing lethality vary from 250 to 600 μg/kg [79,80]. To date, in vitro potencies are reported as AZA-2 > AZA-6 > AZA-34 ≈ 37-*epi*-AZA-1 > AZA-8 ≈ AZA-3 > AZA-1 > AZA-4 ≈ AZA-9 > AZA-5 ≈ AZA-10 > AZA-33 > AZA-26 [81,82].

After the first reported poisoning episode of AZAs in the Netherlands, AZAs have been recorded in molluscs from other European countries (France, UK, Denmark, Spain, and Portugal) and also in countries from Africa, Australia, Asia, South America, and North America including Morocco, China, Chile, Argentina, Canada, and USA [21,83,84]. Recently, the first detection of AZAs in Mediterranean mussels *Mytilus galloprovincialis* [85] was reported. In Europe, it seems that the regulated toxins AZA-1 and AZA-2 are the most abundant analogues from the AZA group, although they occur in different proportions. For instance, in British bivalves, AZA-1 is the most dominant AZA with a ratio AZA-1 to AZA-2 of 2:1 [86]. In the Iberian Peninsula, the dominant toxin is AZA-2, followed by AZA-1 with concentrations below 3 mg/kg of bivalve meat [6,87]. AZA-2 was also the main analogue in bivalves collected on the northern coast of Portugal, followed by AZA-1 and AZA-3 (trace amounts) [88], all below 6.1 mg/kg. Finally, AZA-2 was also the recently described analogue in mussels from the Italian coast [85]. However, some other emerging AZAs are being detected in European waters which include AZA-4, AZA-5, AZA-6 AZA-11, and AZA-43, although in small amounts [87,89]. Another report suggests that AZA-36 and AZA-37 should be included in shellfish safety monitoring programs [81]. In Andalusia, an AZAs profile containing AZA-2 as a predominant analogue has been described and smaller amounts of AZA-43 and AZA-43 disobaric have been attributed to *Amphidoma languida* [87]. 

The main reported vector of Azaspiracid Shellfish poisoning (AZP) is the blue mussel *Mytilus edulis*, although they also have been identified in other species like *M. chilensis* and *M. galloprovincialis* as well as marine sponges, clams (*Dosinia ponderosa*, *Tawera gavi)*, scallops (*Pecten maximus* and *Argopecten purpuratus*), crustaceans (*C. pagurus*), oysters (*Ostrea edulis* and *C. gigas*), and the pen shell *Atrina maura* [48,63,90,91]. Recently, 19 new vectors for AZAs have been reported in three different phyla, including molluscs (*P. ordinaria*, *P. aspera*, *A. depilans*, *S. haemostoma*, *U. umbraculum*, *H. tuberculata*, *P. lineatus*, *G. umbilicalis*, *C. vulgatum*, *C. lampas*), arthropods (*P. pollicipes*), and echinoderms (*P. lividus*, *A. aranciacus*, *O. ophidianus*, *M. glacialis*, *A. lixula*, *S. granularis*, *E. sepositus*, *D. africanum*) [92].

### 2.3. Palytoxins

Palitoxin (PLTX) is the largest and most potent non-peptide toxin identified to date [93]. It is mainly produced by coral anemones of the genus *Palythoa* (*P. tuberculosa*, *P. toxica*, *P. vestitas*, *P. craibdea*, *P. mamillosa*) and by the dinoflagellate *Ostreopsis ovata* [94]. Structurally, PLTX is a large, complex molecule with a long polyhydroxylated and partially unsaturated aliphatic backbone, with more than 100 carbons with 64 chiral centers (Figure 2D). It was first isolated from a *Palythoa* species from Hawaii in the early 1970s [95]. Afterwards, several analogues including homopalytoxin, bishomopalytoxin, neopalytoxin, deoxypalytoxin, and 42-hydroxy-palytoxin were subsequently identified in the *Palythoa *species [96]. Ostreocin-D, ovatoxin-a, -b, -c, -d, and -e, as well as mascarenotoxin-a, -b, and -c, were identified in the benthic dinoflagellates of the genus *Ostreopsis* [97]. All the mentioned toxins can block the Na^+^/K^+^-ATPase pump and exert their potent biological activity by altering normal ion homeostasis in excitable and non-excitable tissues [93]. PLTX accumulates in numerous organisms such as corals, sponges, mussels, and crustaceans [98]. The precise number of species that are susceptible to PLTX accumulation is yet to be known but it is thought to range between 300 to possibly 400, globally. Marine organisms contaminated with PLTX appear to have a bitter and metallic taste, which prevents consumers from ingesting large amounts. Despite this, due to its high toxicity, human fatalities have been well documented [94,99,100,101,102]. In the last two decades, *Ostreopsis* spp. has caused relevant negative impacts on human health through contaminated seafood and dermal contact in the Mediterranean Sea (Spain, Italy, and France), causing respiratory distress and skin irritation in swimmers [100]. One of the highest toxins amounts ever recorded occurred on the coast of France with a total of 0.39 mg for the sum of OVTX-a and PLTX per kg of digestive tube of the flathead mullet *Mugil cephalus* [103]. On the coast of Genova (Italy), several hundred persons had to be hospitalized after exposure to aerosols during a bloom of *Ostreopsis* sp. in the summer of 2005 [99]. In Almeria (Spain), an epidemic outbreak proceeded with respiratory symptoms was reported in 2006 due to toxic microalgae exposure [101]. In France, between 2006 and 2009, a total of nine blooms were registered on the Mediterranean coast in which a total of 47 patients presented symptoms of respiratory irritation and an 8-year-old girl required hospitalization because of the dyspnea caused by extensive rhinorrhea and bronchorrhea [94]. In addition, cases of respiratory problems and skin irritations in humans associated with massive blooms of *O. ovata* in Croatian waters were reported for the first time in the northern Adriatic Sea in 2010 [99]. It is difficult to assess the risk of PLTX poisoning through shellfish consumption due to their co-occurrence with other marine toxins. In 2005 the European Union Reference Laboratory for Marine Biotoxins (EU-RLMB) set a provisional limit of 250 μg/kg of PLTX in shellfish [104]. Later on, EFSA suggested decreasing the limit to 30 μg/kg of the sum of PLTX and ostreocine-D in meat [30]. However, the occurrence of PLTX in foodstuff is not regulated in the EU and there is no recognized official method for the determination of PLTX-group toxins. EFSA expressed concern and demanded assessment of the chronic toxicity of this potent marine toxin. Initial studies on the chronic PLTX toxicity after repeated daily oral administration of PLTX to mice led researchers to determine a no-observed-adverse-effect level (NOAEL) of 3 µg/kg/day for a 7-day exposure period [105]. Recently, the chronic toxicity of PLTX was evaluated after oral administration to mice by gavage during a 28-day. A lethal dose 50 (LD50) of 0.44 µg/kg of PLTX and a NOAEL of 0.03 µg/kg for repeated daily oral administration of PLTX were fixed [106].

### 2.4. Ciguatoxins

Ciguatoxins (CTXs) are complex polyethers composed of 13–14 rings fused by ether linkages that exert their mode of action by activating voltage-gated sodium channels (Na_v_) on cellular membranes, leading to an increase of permeability to sodium ions and cell disruption (Figure 2F) [107,108]. CTXs are lipid-soluble heat-stable compounds, with no odor or taste, causative of the most prevalent seafood born illness worldwide, ciguatera poisoning (CP) [109]. This syndrome is characterized by an acute and a chronic stage, with approximately 175 different symptoms having been described, to date. Symptomatology englobes neurological disturbance (cold allodynia, paresthesia, dysesthesia, sensory hindering like myalgia, pruritus, metallic taste, hyperesthesias, cold allodynia), cardiovascular derangement (hypotension and bradycardia), and gastrointestinal distress (vomiting, abdominal pain, diarrhea) [110]. Geographically, this group of biotoxins is circumscribed to the latitudes of 35° N and 35° S, being considered endemic in the Caribbean (C-CTX), Indic (I-CTX), and South Pacific (P-CTX) regions, though in the past two decades it has also been reported in more temperate regions [111,112,113,114,115,116]. Regarding origin, CTXs have been linked to dinoflagellates from the genera *Gambierdiscus* and *Fukuyoa*, that grow in sediments, attached to seaweeds and coral reefs in tropical and subtropical shallow waters. Consequently, blooms of these microalgae are discreet, due to their epibenthic nature, making their monitoring and managing a challenging task [117].

CTXs result from the biotransformation in herbivorous fish of their precursor gambiertoxins, leading to more toxic forms along the food chain [118]. In this sense, top predators are more prompt to be highly toxic (families: Muraenidae, Serranidae, Sphyraenidae, Lutjanidae), yet these biotoxins have also been detected in detritivores invertebrates [119,120]. As already mentioned, CP is the most common type of intoxication syndrome, even beyond the endemic areas, due to a large number of fish exports, which is estimated at 10,000 to 50,000 intoxications per year worldwide [110,112]. In the EU, the first case of intoxication was reported in 2004, in the Canary Islands, Spain, after the consumption of 26-kg amberjack (*Seriola rivoliana*), leading to the hospitalization of five persons that exhibited CP symptoms that, in some cases, persisted for months: cardiovascular (bradycardia—two persons), systemic (fatigue—five persons, itching—three persons, dizziness—one person), and neurologic distress (myalgia—three persons, paresthesia—three persons, paresthesia—two persons, and reversal of hot and cold sensations—three persons) gastrointestinal (diarrhea—four persons, nausea—three persons, sensory hindering/metallic taste—one person). It was determined that the toxin responsible for this intoxication was 1.0 ppb (ng/g). C-CTX-1 [121]. Other poisoning cases followed (Table 1), with 34 ciguatera outbreaks being registered, englobing 209 poisoning cases in Spain, Portugal, France, and Germany, between 2012 and 2019. Neurological symptoms were present in every outbreak; gastrointestinal symptoms appeared in the majority of the cases, while cardiovascular symptoms were reported in a lesser number. The outbreaks in Spain and Portugal are due to the consumption of autochthonous fish mainly *Seriola* and *Epinephelus* genus and France and Germany reported cases because of consumption of imported fish mainly from the genus *Lutjanus* [31].

Concerning regulatory levels, the United States Food and Drug Administration (FDA) entrenched a guidance concentration of 0.01 μg P-CTX-1B equivalents/kg of tissue and 0.1 C-CTX-1 equivalents/kg of tissue as expected to not exert effects in consumers [122]. This value was based on the toxicity equivalency factors (TEFs), determined by the CTX values acute intraperitoneal LD_50_ in mice as follows: P-CTX-1 = 1, 51-hydroxy P-CTX-3C = 1, P-CTX-3 = 0.3, P-CTX-2 = 0.3, C-CTX-2 = 0.3, P-CTX-3C = 0.2, C-CTX-1 = 0.1, P-CTX-4A = 0.1, 2,3-dihydroxy PCTX-3C = 0.1, and P-CTX-4B = 0.05 [11]. In Europe, consumers are protected by commission regulation (EC) nr 854/2004, which mandates that fishery products containing CTX are forbidden to enter the market [123]. Nevertheless, there is a need for monitoring strategies and the determination of limit values regarding this biotoxin group.

**Table 1 marinedrugs-20-00199-t001:** Some human incidents due to emerging marine toxins in the last 15 years.

Toxin	Report Location	Year	Vector/Uptake Route	Incident	No. Poisonings	Refs.
** *Imine Cyclic* **						
PnTX-G	Ingril Lagoon (France)	2010	Mussels (*Mytilus galloprovincialis*) and clams (*Venerupis decussata*)	1200 mg/kg of PnTX-G in mussels and clams	0	[60]
** *Azaspiracids* **						
AZAs	Norway	2005	Viscera of the edible (brown) crab, *C. pagurus*,	Hospitalization of 2 persons after eating crabs containing AZA.	2	[7]
AZAs	Coast of Sweden	2018	*Azadinium* spp.	AZA levels above the regulatory limit	0	[7]
AZAs	North Sea coast, Netherlands,	2020	*Phaeocystis globosa*	Human fatalities: 5 persons playing water sports died.	5	[7]
** *Palytoxins* **						
** *PlTXs* **	Genova (Italy)	2005	*Ostreopsis* sp./exposure to aerosols	Hospitalization of several hundred persons.	>100	[99]
***PlTX*, *ovatoxin-a***	Ligurian Coasts	2006	*O. ovata*	Human toxic outbreak. Bathing was forbidden	Few cases	[100]
** *PlTXs* **	Almeria (Spain)	2006	*Ostreopsis* spp./exposure to aerosols	Epidemic outbreak with respiratory symptoms	>100	[101]
** *PlTXs* **	French Mediterranean coast	2006–2009	9 blooms *Ostreopsis* spp.	Respiratory irritation in 47 swimmers. Hospitalization of an 8-year-old girl (dyspnea).	48	[94]
** *Ciguatoxins* **						
** *CTXs* **	Madeira archipelago, Portugal	2007–2008	No vectors were identified	Hospitalization of 6 persons exhibiting CP symptomatology	6	[124]
** *CTXs* **	Madeira archipelago, Portugal	2008	Amberkacl (*Seriola* spp.)	Hospitalization of 11 persons after consumption of the contaminated fish (CTX concentration NDA)	11	[124]
** *CTXs* **	Spain	2012	Amberjack (*Seriola* spp.) and Grouper (*Epinephelus* sp.)	Poisoning victims with symptoms consistent with CP, after ingestion of a predatory local fish; 12 intoxications were confirmed analytically for CTX.	37	[31]
** *CTXs* **	Portugal	2012	Amberjack and Barred Hogfish (*Seriola* sp. *Bodianus scrofa*)	Hospitalization of 12 poisoning victims, CTX NDA	12	[31]
** *CTXs* **	Spain	2013	Grouper (*Epinephelus* sp.)	Poisoning victims with symptoms consistent with CP, intoxications were confirmed analytically for CTX	15	[31]
** *CTXs* **	Spain	2015	Grouper (*Epinephelus* sp., *Mycteroperca fusca*) and Bluefish (*Pomatomus saltatrix*)	Poisoning victims with symptoms consistent with CP, 2 intoxications were confirmed analytically for CTX	8	[31]
** *CTXs* **	Portugal	2015	Grouper (*Epinephelus marginatus*)	Hospitalization of 4 out of 7 poisoning victims, CTX NDA	7	[31]
** *CTXs* **	Spain	2016	Grouper and Amberjack (*Epinephelus* sp. and *Seriola* sp.)	Poisoning victims with symptoms consistent with CP, intoxications were confirmed analytically for CTX	5	[31]
** *CTXs* **	Portugal	2016	Red Porgy (*Pagrus pagrus*)	Poisoning victims with symptoms consistent with CP, intoxications were confirmed analytically for CTX	4	[31]
** *CTXs* **	Spain	2017	Grouper (*Epinephelus* sp., *Mycteroperca fusca*)	Poisoning victims with symptoms consistent with CP, intoxications were confirmed analytically for CTX	2	[31]
** *CTXs* **	Spain	2018	Triggerfish (*Canthidermis sufflame)*	Hospitalization of 1 person. Poisoning victims with symptoms consistent with CP. CTX NDA	4	[31]
** *CTXs* **	Spain	2019	Amberkacl (*Seriola* spp.)	Poisoning victims with symptoms consistent with CP, intoxications were confirmed analytically for CTX	6	[31]
** *Tetrodotoxins* **						
*TTX* and 5,6,11-trideoxyTTX	Spain	2008	Trumpet Shell (*Charonia lampas*)	Hospitalization of a person who ate a contaminated gastropod (315 mg TTX/kg)	1	[22]

Definitions: AZA: azaspiracid. PnTX-G: Pinnatoxin-G. PlTXs: palitoxins. CTX: ciguatoxins. NDA: not determined analytically.

### 2.5. Brevetoxins

Brevetoxins (BTXs) are lipophilic neurotoxins mainly produced by the dinoflagellate *Karenia brevis* [125]. These cyclic polyethers are grouped into two principal chemical forms based on their backbone, type A and B (Figure 2B). To date, approx. 70 BTX derivatives have been described. For most of them, the metabolizing products in shellfish are of the two main parental toxins, BTX-1 (type A) and BTX-2 (type B) [32,126]. BTXs are responsible for the neurotoxin shellfish poisoning (NSP) syndrome, being the most common via of intoxication through ingestion, aerosol inhalation, or dermal contact [127,128,129]. Symptoms produced by exposure to these neurotoxic toxins range from nausea, vomiting, diarrhea, respiratory tract irritation, rhinorrhea, burning sensation in the nose and throat, bronchoconstriction, paraesthesia, dizziness, loss of coordination, cramps, paralysis, seizures, and, in severe cases, coma [32,127,128]. Poisoning features usually manifest between one hour to 24 h after ingestion, and still no antidote is available [130]. The action mechanism of BTXs occurs by their binding specifically to site-5 of Na_v_s, leading to the persistent activation of the channel, producing the influx of sodium ions and depolarizing neuronal and muscle membranes [131]. Some studies also point to the potential capacity of this group of toxins to induce DNA damage and chromosomal aberrations, and their chronic toxicity remains to unravel [28].

BTXs are considered endemic to the areas of the Gulf of Mexico, Florida, West Indies, and New Zealand, where *K. brevis* blooms are more prevalent, although with no recorded fatalities ever [132]. For consumers’ protection, these countries established the limit value of 0.8 mg BTX-2 equivalents/kg of shellfish as safe [32]. In the EU BTXs still have no regulatory status, due to the scarcity of animal toxicity and human illness quantitative data regarding this group, which hampers the calculation and establishment of tolerable daily intake (TDI) and acute reference dose (ARfD) [32]. Recently the first report of BTXs in Europe was released, and these toxins were detected for the first time in mussels in the French island of Corsica during the winter season. No fatalities or poisoning incidents were recorded. Determined quantities ranged between 82 and 344.6 µg BTX-2 + BTX-3/kg digestive gland. BTXs were detected in the autumn and winter of 2018, but retroactive analysis of conserved mussel samples disclosed the presence of these neurotoxins, in the same site, in November of 2015 (only BTX-3) [132]. Regarding producers, although the presence of *K. brevis* was not recorded, other *Karenia* species were detected in the Diana lagoon by Amzil and colleagues: one unidentified *Karenia* species, *K. mikimotoi*, *K. papilionaceae,* and *K. longicanalis*. Additionally, two raphidophytes suspected to be involved in BTXs production, *Heterosigma akashiwo* and *Fibrocapsa japonica*, were detected [132,133]. As a mitigation measure, a workgroup was set by the French Agency for Food, Environmental, and Occupational Health and Safety (Anses). As a result of this work, a guidance level of 180 µg BTX-3 eq./kg shellfish meat considering a protective standard portion size of 400 g shellfish meat was proposed. Additionally, two lowest-observed adverse effect levels (LOAELs) were calculated based on the available information regarding the human intoxication reports [134]. With this confirmed emergent challenge, it is pertinent to gather data to establish legislation and monitoring procedures regarding this group of toxins.

### 2.6. Tetrodotoxins

Tetrodotoxin (TTX) is a potent neurotoxin, with a molecular weight of 320.11 g/M, and a chemical formula of C_11_H_17_N_3_O_8_ (Figure 2E). Structurally, this alkaloid is characterized by a guanidinium moiety, a pyridine ring with additional fused ring systems, and six hydroxyl groups, weakly basic and positively charged at a physiological pH [135]. TTX was first discovered in 1909 by Tahara and Hirata, being isolated from puffer fish ovaries [136]. Initially, this neurotoxin was only associated with the Tetraodontidae family and the Pacific area; however, to date, a diverse range of aquatic and terrestrial organisms have been reported as TTX bearers worldwide (chaetognaths, platyhelminthes, nematodes, molluscs, arthropods, echinoderms, fish, newts, and frogs) [137]. The reason behind TTX’s broad distribution, especially in the marine environment, is due to its exogenous origin, bacteria, present in the sediment or associated with their hosts [138,139]. TTX most described producers belong to the genus *Vibrio*, *Pseudomonas*, *Bacillus*, *Shewanella*, *Nocardiopsis*, *Alteromonas,* and *Roseobacter* [138,139,140].

To date, approximately 30 TTX analogues have been described, with their degree of toxicity being determined by the occurrence of structural changes at carbons C-6 and C-11, implying their greater or lesser affinity with their molecular target, the Na_v_. TTX exerts its action by occluding site one of the outer pore of Na_v_, inhibiting cellular communication, hindering the generation of action potential and impulse conduction. Acute symptoms can vary from paraesthesia, perioral numbness, incoordination, early motor paralysis, hypotension, bradycardia, and unconsciousness, culminating in death by cardio-respiratory failure [33,141,142]. Without a known antidote, the only mitigation measures available to combat a TTX poisoning incident are ventilatory support and gastric lavage [143]. In terms of toxicity, this group is approximately a thousand times more toxic compared to cyanide, being heat stable and water-soluble, and cooking processes can enhance the level of toxicity within contaminated food items [144,145]. Past knowledge described that TTX and its analogues were endemic and circumscribed to the Asian region, but in the last two decades, this group has expanded into more temperate ecosystems. It is believed that the increase in average water temperature, caused by climate change, together with the opening of new maritime corridors and anthropogenic inputs have been key factors in this emerging phenomenon [22,24,26,143,146]. From 2008 to date, TTX reports in the EU region have become more frequent, from more warm waters (Portugal, Spain, Italy, and Greece) to more northern regions (France, Netherlands, United Kingdom, and Ireland). TTX was detected in several species of bivalve molluscs (*Mytillus galloprovincialis*, *Donax trunculus*, *Crassostrea gigas*, *Spisula solida*, *Pecten maximus*), gastropods (*Phorcus lineatus*, *Gibulla umbilicalis*, *Patella depressa*, *Nucella lapillus*), and echinoderms (*Echinus esculentus* and *Ophidiaster ophidianus*), and the majority of positive hits in samples were collected in areas of water temperature above 15 °C [22,23,25,26,147,148,149,150].

Concerning consumers’ protection, until 2017 only regulations 853/2004/EC and 854/2004/EC, published in 2004, preventing fish species from the families of Tetraodontidae, Canthigasteridae, Molidae, and Diodontidae, reported as TTX-bearers, from market placement were in force [123,151]. Later, as a result of the inclusion of TTX in the Dutch monitoring program, in March 2017 EFSA delivered a scientific opinion concerning TTX group presence in marine bivalves and gastropods [33]. In this document, EFSA proposes the limit of 44 µg TTX equivalents/kg shellfish meat, considering it safe based on the no-observed-adverse-effect level (NOAEL) of 75 µg/kg body weight [33]. Additionally, the organization recognizes the need for a more tight scientific dialogue, appealing to the gathering of more data concerning: (i) TTX occurrence in edible bivalves and gastropods from EU territory; (ii) the need for certified standards and reference materials; (iii) quantification data using EU approved and validated chemical-analytical methods, (iv) fate and stability studies; (v) studies to unravel the sources and critical factors that lead to accumulation in marine bivalve molluscs and gastropods; (vi) toxicokinetics of TTX and its derivatives; (vii) additional data on oral acute and chronic toxicity of the TTX group; (viii) relative potencies of TTX and its derivatives, preferably after oral exposure; and (ix) due to Saxitoxin and TTX chemical similarities and mode of action, the possibility of combining both groups in one HBGV [33]. In 2019, Boente-Juncal and collaborators answered EFSA’s recommendations and evaluated the chronic oral toxicity of TTX. This study proved that low oral doses (75 µg/kg) of TTX have detrimental effects on renal and cardiac tissues, detecting alterations in blood biochemistry parameters and urine [152]. In this sense, there is a need for more studies to add up knowledge that can culminate in the proper regulation of this group of toxins to ensure the safety of consumers.

## 3. Challenges for the Detection of Emerging Toxin Detection

### 3.1. Cyclic Imines

The availability of CI standards throughout the world has been a long-term problem due to the difficulty of obtaining them through extraction from molluscs or their challenging synthesis [1]. For this reason, several alternative functional methodologies for CI have been developed in the last decade (Table 2). Otero and co-workers developed a method for the detection and quantification of SPXs in mussel samples using a direct fluoresce polarization (FP) assay with nAChR from *Torpedo marmorata* membranes. The method uses receptors from *T. marmorata* membranes labelled with a derivative of fluorescein. This assay is a reproducible, simple, and very sensitive direct method useful for quantifying SPX-13 in the range of 50–350 µg/kg shellfish meat [50]. Simultaneously, the same group developed another FP assay using the same membranes, but based on the competition of SPXs with a-bungarotoxin for binding to nACh [42] and, a chemiluminescence method based on the competition between toxins and biotin-α-bungarotoxin immobilized on a streptavidin-coated surface for binding to nAChR [153]. In parallel, a solid-phase receptor-based method for the detection of CI using a microsphere-flow cytometry system (Luminex) was developed. The method allows the detection of the SPXs in mussels, clams and scallops, in the range of 10–6000 μg/kg of shellfish meat and a LOD of 3 μg/kg [154]. Araóz and colleagues developed a microplate-receptor binding assay for the detection and identification of neurotoxins based on the high affinity of the toxins for their receptor targets. The assay is commercialized by ABRAXIS [155] and it is sold as a high throughput method for rapid detection of nicotinic neurotoxins directly in environmental samples. The method is suitable for monitoring nanomolar concentrations of CI in drinking, surface, and groundwater as well as in shellfish extracts. In parallel, the same group have released a new rapid and quantitative method for the detection of CI based on a lateral flow assay [156]. However, given that imine cyclic toxins are lipophilic toxins susceptible to detection using the official chromatographic method for regulated ones, some CIs for which standards are available (SPX-13, SPX-13,19, and PnTXG) are currently reported by the multitoxin detection method for the determination of lipophilic marine toxins by LC-MS/MS. In addition, given the potential impact on human health, several MS/MS methods based on fragmentation pathways of reference toxins (SPX-13, PntX-G) have been proposed to study their occurrence [48].

### 3.2. Azaspiracids

The development of methods for AZAs has been limited due to few purified toxins. AZA-1 have been successfully synthesized since 2008 [172]. Nevertheless, the chemical synthesis of these compounds to develop standards is challenging and complex [173]. Some AZAs such as AZA-3 and AZA-6 can only be provided from contaminated molluscs so that the purification from marine organisms seems inevitable [174,175]. Regarding detection methods, research has been focused on the production of monoclonal and polyclonal antibodies against AZA [159,176,177]. These have been developed into a competitive enzyme-linked immunosorbent assay (ELISA) [157], a magnetic bead/electrochemical immunoassay [158], and an immunosensor [178] for the polyclonal antibodies. The monoclonal antibodies were also used by the different research laboratories to develop an immunoassay for the detection of these biotoxins such as a microsphere/flow fluorimetry-based immunoassay [179,180]. Despite the handicap of standard availability, the analytical methodology based on LC-MS/MS seems the best approach. When the LC-MS/MS methodology is used for the detection of emerging AZAs, the official method for the detection of lipophilic toxins is often performed. Here are included the MS transitions for a wider range of AZAs for which standards are not available. Blanco and co-workers developed a method for the identification of emerging AZAs by LC-MS/MS in bivalve molluscs [87]. Intending to check if emerging AZAs were present in commercial mussels, Otero and colleagues monitored a total of 44 analogues using the same chromatographic conditions as those used for regulated AZAs, including the specific transitions in the MS method [6].

### 3.3. Palytoxins

The identification and quantification of PLTXs in seafood, fish, and coastal waters are nowadays of paramount importance to prevent consumers from serious intoxication incidents. Currently, there is no reference method for the detection and quantification of these toxins. Mouse or rat bioassays have been traditionally utilized for their monitoring in seafood; however, these methodologies are currently being replaced by methods based on functional assays and chemical methods. Several detection methods based on the interaction between the Na, K-ATPase, and the PLTXs were developed [104]. PLTXs can be quantified by FP in the nM range labelling the Na, K-ATPase with a reactive succinimidyl ester of carboxyfluorescein and, measuring the FP of protein-dye conjugate when the amount of PLTX in the medium is modified [160]. Zamolo and co-workers have developed a sandwich immunoassay and electrochemiluminescence method to detect PLTX in which the electrochemiluminescence is directly proportional to PTX concentration with an LOD in both mussel and algal samples of 220 ng/mL [162]. Fraga et al. describe an immuno-detection method for PLTX-like molecules based on the use of microspheres coupled to flow-cytometry detection [161]. The assay consists of the competition between free PLTX-like compounds in solution and PLTX immobilized on the surface of microspheres for binding to a specific monoclonal anti-PLTX antibody. The assay displays a dynamic range of 0.47–6.54 nM for PLTX and is suitable for mussel samples in a range of 374 ± 81 to 4430 ± 150 μg/kg [161]. Several alternative chemical methods for the identification and characterization of PTLXs-like molecules in marine organisms have been also released (Table 3) [161,181].

### 3.4. Ciguatoxins

Despite the high prevalence CP worldwide, the under-reporting of poisoning cases and inadequacy or fragmentation of sources of information regarding this emerging phenomenon poses a problem in terms of efficiency of gathering valuable data to build legal and scientific documents to better serve consumers [112,182]. More than 30 analogues have been described to date, though the lack of reference materials and calibration standards hinders the development of efficient detection methods and mechanistic and pathological studies [183].

For research and CTX determination purposes, the application of a two-phased protocol is advised, using a semi-quantitative bioassay (receptor-binding assays and in vitro Neuro-2a cell cytotoxicity assay) paired with a chemical analytic method for confirmation (LC-MS/MS) (Table 2 and Table 3) [110,166,184]. The receptor-binding assays are based on the use of a preparation of CTXs molecular target, brain membrane composition rich in Na_v_, where the biotoxin present in a sample competes with a radiolabeled toxin to bind to their molecular target [166]. Although the method offers high sensitivity and specificity and can quantify the toxic potency of a sample, there is still a need for interlaboratory validation. The in vitro Neuro-2a cell cytotoxicity assay (N2a-assay) is a cytotoxicity method, derived from the use of neuroblastoma cell line, obtained from mice (*Mus musculus*). In resemblance with the receptor-binding assay, the N2a-assay specifically detects compounds that act on Na_v_s, being proved to qualitatively detect CTX in samples, and estimate their toxicity. One of the major drawbacks of the method is the use of ouabain (O) and veratridine (V) to increase the assay specificity and sensitivity since these compounds are toxic to neuroblastoma cells, masking positive results. To overcome this matter, Loeffler and colleagues proposed the concentration of 0.22/0.022 mM O/V as optimal to obtain safe and reliable results for the detection of CTX-3C and CTX-1B [163]. Still, there is a need for inter-laboratory consensus and validation. Regarding chemical-analytical methods, in 2018 Shibat and colleagues presented a detection method using different matrices: phytoplankton (*Gambierdiscus polynesiensis*), echinoderms (*Tripneustes gratilla*), gastropods (*Tectus niloticus*), and fish (*Chlorurus microrhinos* and *Epinephelus polyphekadion*) [185]. Here, two methods (low and high-resolution mass spectrometry (LC-LRMS or HRMS)) for the identification and quantification of Pacific CTX were tested. Although the LRMS method showed better sensitivity, HRMS allowed the identification of artefacts and was indicated as a good tool for confirmation of the identity of P-CTXs analogues. Hence, both methods were considered complementary for the identification and quantification of P-CTXs [185]. In March of 2021, EFSA released an external scientific report proposing two different, but complementary, LC-MS/MS approaches to detect and quantify CTXs (Table 3) [15]. Here the extraction and purification steps were optimized, and the analysis results demonstrated adequate levels of detection and quantification, in line with the literature. Additionally, it is recommended that in the absence of CTX standards or reference materials, the monitoring of water losses and characteristics fragments typical of this biotoxin group can be used as a contingency for monitoring purposes. Nevertheless, a confirmation using HRMS (High-Resolution Mass Spectrometry) is required. HRMS is a powerful confirmation tool, yet it needs higher sample amounts and extra clean-up steps due to its lower sensitivity [186].

### 3.5. Brevetoxins

For the detection and quantitation of Brevetoxins, in vitro and immunoassays have been proven effective to detect this biotoxin group in marine matrices (Table 2). However, hyphenated techniques, like the LC-MS/MS, present themselves as valuable tools for the determination and quantitation of BTXs due to the high degree of specificity delivered (Table 3). Nevertheless, the validation of such methods depends on the availability of standards and reference materials. In 2012, MacNabb and co-workers published a single laboratory validation for BTXs (BTX-B1, BTX-B2, S-deoxyBTX-B2, BTX-B5, BTX-2, and BTX-3) in four different shellfish matrices (*Crassostrea virginica*—eastern oyster, *Perna canaliculus*—green shell mussel, *Mercenaria mercenaria*—hard clam, *Crassostrea giga*—Pacific oyster). The developed method showed good sensitivity, originating the regulatory limit of 0.8 mg/kg BTX-2 equivalents in the New Zealand BTX endemic area [187]. As described in Section 3.4, Amzil and colleagues used for the determination of BTXs, a multi-toxin LC-MS/MS approach was used efficiently (Table 3) [132]. Nevertheless, as a final remark, a two-phased screening for this biotoxin group was also recommended: first with an immunoassay for a broad BTX detection, followed by chromatographic confirmation [134].

### 3.6. Tetrodotoxins

TTX group is a challenging one due to the lack of data in the EU to assemble proper information to elaborate and establish effective legislation respecting food safety. Regarding methods of detection and quantification of this group, in the EU there are three main types of methodologies used for TTX determination: cell-based assays, antibody-based methods, and chemical-analytical methods. Cell-based assays are high-throughput and rely on the action mechanism of the target toxin group being able to assess the toxicity of TTX analogues [171]. On the other hand, they cannot provide information regarding the toxin profile nor discriminate between Saxitoxins from TTX as they share the same mode of action. Antibody-based methods are advantageous as a rapid qualitative pre-screening, allowing the estimation of the concentration within antibody cross-reactivity. The weaknesses of the method are due to fact that it only detects the presence of the toxins that the antibody cross-reactivity allows (overall toxicity cannot be estimated) and the incapacity to provide information on the toxin profile. Chemical-analytical methods present themselves as the most suitable for TTX screening and analysis since they provide information on the toxin profile being able to separate, identify, and quantify TTX and its analogues. Highlighting the most recently developed methods, we have the method by Rodriguez et al. (2018). This method is characterized by being a multitoxin detection method, being able to detect TTX and 5 analogues [188]. Recently, in 2021, the EU reference laboratory for marine toxins published the single laboratory validation, intending to organize a future intercomparison study with interested member states [189]. Both methods have similar sensitivities (Table 3), the latter being able to detect an additional analogue, 6,11-dideoxy-TTX [189]. Nevertheless, they are incapable of giving toxicity information and are dependent on available toxic equivalent factors (TEFs), standards, and reference materials. Still, antibody and cell-based methods require a chromatographic confirmation, and although validated within laboratories, all methods require further interlaboratory validation [33].

**Table 3 marinedrugs-20-00199-t003:** HPLC based methodology recently developed for the identification of emerging marine toxins.

Equipment	Chromatographic Column	Mobile Phase	Toxins Tested	LOD/LOQ	Refs.
** *Imine Cyclic* **					
UPLC-MS/MS	Aquity UPLC BEH C18 (2.1 µM × 100 mm, 1.7 µm, Waters, Barcelona, Spain)	A = 100% water. B = acetonitrile:water (95:5), both containing 50 mM FA and 2 Mm AM.	PnTX-A,B,C,D,E,F,G GYM-A,B,C,D, 12-Me GYM-A. SPX-13, SPX-13,19 SPX-A,B,C,D,E,F,G,H,I. 20-MeSPX C. 27-OH-13,19-didesMe SPX-C. 27-OH-13-desMe SPX-C.	0.1 µg/kg SPX-13, SPX-13,19, and PnTX-G.	[48]
LC-MS/MS	Agilent ZORBAX SB-octylsilyl (C8) (50 × 2.1 mm id, 1.8 μm).	A = 100% water B = acetonitrile:water (95:5), both with 2 mM AF and 50 mM FA.	PnTX-G. GYM-A. SPX-13	LOD = 0.3 µg/kg and LOQ = 1 µg/kg.	[46]
** *Azaspiracids* **					
LC-MS/MS	Chromatographic column Acquity UPLC BEH C18 (2.1 × 100 mm, 1.7 µm, Waters).	A = 6.7 mM NH_4_OH (pH11). B = MeCN 90% with 6.7 mM NH4OH.	AZA-1,2,3,4,5,6; AZA-7,8,9,10,11,12; AZA-33,34,35,36,37,38; AZA-39,40,41,42,43; AZA-54,55,56,57,58; Me-AZA2; AZA2 phosphate; AZA11phosphate	LOQ = 42 mg AZA-2/kg meat.	[87]
** *Palytoxins* **					
UPLC-IT-TOF	HSS T3 column. Mobile phases. Temp: 35 °C	A = water. B = acetonitrile. Both acidified with 30 mM FA.	PLTX (*m*/*z* 906.81 and 1359.71) and 42-OH-PLTX (*m*/*z* 912.15 and 1367.72).	LOD = 190 ng/mL. LOQ = 650 ng/mL	[161]
LC-MS/LC-HRMS	Poroshell 120 EC-C18, 2.1 um × 100 mm. 25 ° C.	A = water. B = acetonitrile-water (95:5). Both containing 30 mM AA.	PLTX (*m*/*z* 906.8) and PLTX methyl ester (*m*/*z* 869.4).	LOD = 15 ng/mL	[190]
** *Ciguatoxins* **					
LC-MS/MS (detection and quantification)	Poroshell 120 EC-C18 column (3.0 × 50 mm, 2.7 µm, Agilent)	A = 0.1% FA and 5 mM AF. B = MeOH 0.1% FA and 5 mM AF	CTX-1B, C-CTX-1, 2,3-dihydroxiCTX-3C, 51-hydroxiCTX-3C, 52-*epi*-54-deoxyCTX-1B/54-deoxyCTX-1B, 49-*epi*CTX3C/CTX3C, CTX4A/CTX4B	0.0045 μg/kg	[186]
LC-MS/MS (confirmatory for CTX-C)	Poroshell 120 EC-C18 column (3.0 × 50 mm, 2.7 µm, Agilent)	A = 0.1% FA and 5 mM AF. B = MeCN 0.1% FA and 5 mM AF	C-CTX-1 is based on three water losses and two confirmatory product ions *m*/*z* 191.1 108.9.	0.0045 μg/kg	[186]
LRMS & HRMS (confirmatory)	C18 Kinetex column 50 × 2.1 mm, 1.7 μm, 100 Å (Phenomenex)	A = 2 mM AF B = 95% MeCN, 2 mM AF and 50 mM FA	P-CTX-3C and P-CTX-1B/Seafood and phytoplankton	P-CTX1B (0.075 μg/kg P-CTX3C (0.10 μg P-CTX1B eq./kg)	[185]
** *Brevetoxins* **					
LC-MS/MS	BDS Hypersil C8 (octylsilyl) HPLC column (3 µm, 50 × 2.1 mm, Thermo Scientific)	A = 50%/2.5% IA. B = 97.5% methanol/2.5% IA C = 30 mM AF and 470 mM FA D = 90% acetonitrile	BTX-B1, BTX-B2, S-deoxyBTX-B2, BTX-B5, BTX-2 and, BTX-3	0.025–0.048 mg/kg	[187]
LC-MS/MS	Kinetex XB-C18 (100 × 2.1 mm), 2.6 µm + pre-column Core-shell, 2.1 mm (Phenomenex)	A = 2 mM AF and 50 mM FA. B = MeOH/water (95:5, *v*/*v*), 2 mM AF and 50 mM FA	BTX-2, BTX-3	23 µg/kg	[132]
** *Tetrodotoxins* **					
HILIC-MS/MS	Waters Acquity UPLC Glycan BEH Amide HILIC Column, 130 Å 1.7 μm, 2.1 × 150	A = 0.015% FA + 0.06% of 25% ammonia B = 70% MeCN + 0.01% FA.	TTX, 4-*epi*-TTX, 5,6,11-trideoxy TTX; 11-nor TTX-6-ol; 4,9-anhydro TTX; 5-deoxy TTX/11-deoxy TTX; 6,11-dideoxy-TTX	0.31 ± 0.12 µg/kg	[189]
UPLC-MS/MS	ACQUITY UPLC BEH Amide (2.1 × 100 mm, 1.7 µm, Waters)	A = 0.1% FA and 10 mM AF. B = 95% MeCN 0.1% FA and 2% 100 mM AF	TTX, 4-*epi*-TTX; 5,6,11-trideoxy TTX; 11-nor TTX-6-ol; 5-deoxy TTX; and 4,9-anhydro TTX	0.25 µg/kg	[188]

Definitions: T: temperature: E: elution. IA: Isopropyl alcohol. FA: Formic acid. AF: ammonium formate. LOD: limit of detection. LOQ: limit of quantification.

## 4. Conclusions

Presently, marine toxins, including emerging ones, are the most crucial challenge for shellfish harvesting and marketing. Vigilance and monitoring of coastal waters and marine species is of great importance. The novel emerging harmful algal species with serious public health impacts must be identified before any negative implications on shellfish consumers arise. If toxic food items reach the market undetected, consumers are at risk of intoxication with varying degrees of severity. The information about emerging toxins is paramount to update the data concerning the risk assessment of these compounds and setting up a regulatory level that protects public health is mandatory. In this sense, besides the already established marine toxins found throughout European waters, emerging non-regulated azaspiracids, cyclic imines, palytoxins, ciguatoxins, tetrodotoxins, and brevetoxins have gained much-deserved attention in recent years. Due to their recurrence and frequency, EFSA has proposed a limit of 400 μg SPXs/kg SM, 44 µg TTX eq/kg SM, 30 μg/kg for the sum of PLTX and ostreocine-D. For CTXs, two methodologic approaches have been proposed for group determination. However, these are only recommendations, and they are not included in the official regulations. As the surfacing of novel toxins may occur simultaneously, the development of a methodology that could be applied to the determination of different toxin groups is crucial, providing a broad toxin profile of the contaminated samples. The scarcity of standard reference materials hinders this progress, and it is vital that it is overcome by the scientific community.

Although the number of poisoning cases reported worldwide is considerable, there is a lack of epidemiological studies and the effects of chronic exposure to these compounds are not yet known. Sensitizing the health community is crucial, as is the establishment of an international network for predicting and signaling blooms and poisoning cases.

An international effort must be made to stimulate close scientific dialogue, bringing the academy closer to health professionals promoting awareness of this emerging phenomenon.

## Figures and Tables

**Figure 1 marinedrugs-20-00199-f001:**
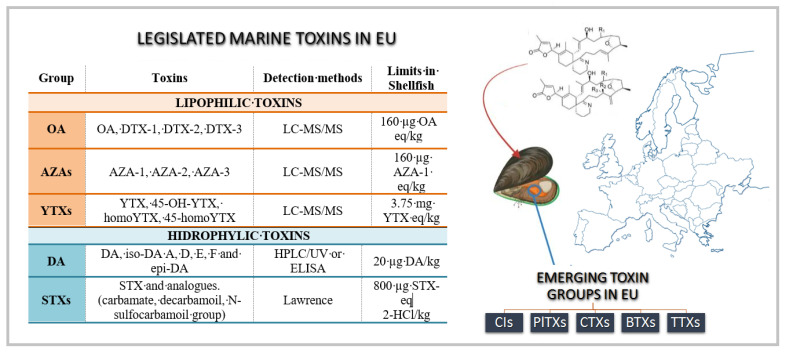
Regulated marine toxins in EU and their maximum levels in shellfish for human consumption: Azaspiracids (AZA), okadaic acid (OA), yessotoxins (YTXs), domoic acid (DA), and saxitoxins (STX). Main emerging marine toxins in EU: cyclic imines (CIs), palitoxins (PlTXs), ciguatoxins (CTXs), brevetoxins (BTXs), and tetrodotoxins (TTXs) [8,9]. Pectenotoxins are not included according to EFSA opinion and novel legislation [10,11,12].

**Figure 2 marinedrugs-20-00199-f002:**
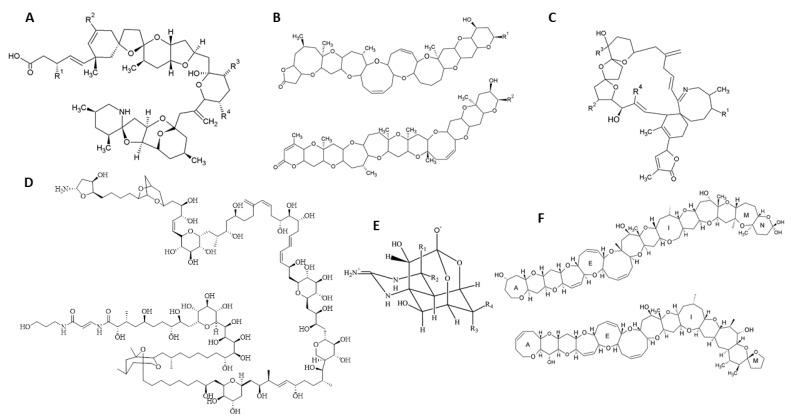
Emerging toxins general structure: (**A**) Azaspiracids, (**B**) Brevetoxins; (**C**) Spirolides; (**D**) Palytoxin; (**E**) Tetrodotoxin; (**F**) Ciguatoxins.

**Table 2 marinedrugs-20-00199-t002:** Recent in vitro methodology for emerging marine toxins identification.

Method	Procedure	Toxins/Matrix Tested	Range or LOQ	Refs.
** *Cyclic Imines* **				
Receptor-based method (FP)	A direct assay based on binding SPXs to nAChRs from *T. marmorata* membranes.	SPX-13/shellfish	50–350 µg SPX-13/kg meat	[50]
Receptor-based method (FP)	Competition between SPXs and a-bungarotoxin for binding to nAChRs.	SPX-13/shellfish	40–200 µg SPX-13/kg meat	[42]
Receptor-based method (Chemiluminescence)	Competition between SPXs and biotin-α-bungarotoxin immobilized on a streptavidin-coated surface, for binding to nAChRs.	SPX-13/shellfish	50 μg SPX-13/kg meat.	[153]
Solid-Phase Receptor-Based Assay (microsphere-flow cytometry system).	Immobilization of nAChR or Ls-AChBP on the surface of carboxylated microspheres and the competition of CIs with biotin-α-BTX for binding to these proteins.	SPX-13/shellfish	10–6000 μg SPX-13/kg of meat and a LOD of 3 μg SPX/kg.	[154]
Non-radioactive Microplate-Receptor Binding Assay (ABRAXIS)	Neurotoxins competitively inhibit biotinylated-α-BTX binding to nAChR in a concentration-dependent manner.	CIs, ATXs	nM range	[155]
Toxin-receptor lateral flow test “NeuroTorp”	Based on the immobilization of nAChR on high porosity borosilicate membrane filter support, and the use of a biotinylated α-BTX as toxin-tracer.	ATX-a and CIs (PnTXs, SPXs, GYM)	nM range	[156]
** *Emerging Azaspiracids* **				
Immunoassay	ELISA. Ovine polyclonal antibodies	AZA1-3 and Emerging AZAs including AZA-4−10, -33, and -34 and 37-epi-AZA-1. AZA-17 and AZA-19.	57 μg/kg shellfish	[157]
Immunoassay	ELISA. Antibody immobilization supports MBs. Tracer: AZA-HRP		63 μg AZA-1 eq./kg)	[158]
Immunoassay	ELISA. Plate-coater: OVA−cdiAZA1.	AZA reference materials as well as the precursors to AZA-3 and AZA-6,	37 μg/kg for AZA-1 in shellfish.	[159]
** *Palytoxins* **				
Receptor based method (FP)	Based on the interaction between the Na, K-ATPase, and PLTX.	PLTX/mussels, and ostreopsis	LOQ = 10 Nm LOD = 2 Nm	[160]
Immunodetection method (microspheres coupled to flow-cytometry detection).	Based on the competition between free PLTXs in solution and PLTX immobilized on the surface of microspheres for binding to a specific monoclonal anti-PLTX antibody.	PLTXs/musels	Dynamic range: 0.47–6.54 Nm and LOQ: 374–4430 μg/kg.	[161]
Electrochemiluminescence method	Electrochemiluminescence is directly proportional to PTX	PLTXs/mussel, algal samples	LOD = 220 ng/mL	[162]
** *Ciguatoxins* **				
Cell-based assay	Sensitivity to neuroblastoma N2a cell line	CTX-3C and CTX-1B/fish flesh	1.35 pg CTX-3C/mL and 2.06 pg CTX-1B/mL	[163]
Cell-based assay	Sensitivity to neuroblastoma N2a cell line	P-CTX-1 eqs/lionfish	0.0039 ppb–0.0096 ppb P-CTX-1 eq.	[164]
Cell-based assay	Sensitivity to neuroblastoma N2a cell line	P-CTX-1/SPATT	0.02 ng P-CTX3C eq./g	[165]
Immunoassay	Radioligand receptor binding assay	P-CTX-3C/fish flesh	0.75 ng P-CTX-3C eq./g	[166]
** *Brevetoxins* **				
Cell-based assay	Sensitivity to neuroblastoma N2a cell line	BTX-3/fish flesh	3.04 ng BTX-3/mL	[163]
Immunoassay	ELISA	BTX-3/clam and oyster	0.04 µg BTX-3 eq./g shellfish	[167]
Immunoassay	Radioligand receptor binding assay	BTX-1, BTX-3, BTX-9/*K. brevis*	1 Pm to 1 µM BTX-2	[168]
** *Tetrodotoxins* **				
Cell-based assay	Sensitivity to neuroblastoma N2a cell line	TTX/shellfish	20 µg TTX/kg	[169]
Immunoassay	Competitive inhibition enzymatic immunoassay (Melisa)	TTX/mussels and oysters	20 µg TTX/kg and 30 µg TTX/kg	[170]
SPR	Nanoarray planar waveguide biosensor	TTX/puffer fish	0.4 to 3.29 mg/kg	[171]

Definitions: α-BTX: α-bungarotoxin. ELISA: enzyme-linked immunosorbent assay. Eq: equivalents. FP: fluorescence polarization. LOD: limit of detection. LOQ: limit of quantification. MB: magnetic bead. NAChRs: acetylcholine receptors. N2a: neuro-2a. OVA: ovalbumin. SPATT: Solid Phase Adsorption Toxin Tracking. SPR: Surface Plasmon Resonance.

## Data Availability

Not applicable.

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
