# Peer review of "Emerging Marine Biotoxins in European Waters: Potential Risks and Analytical Challenges"

_marinedrugs, 2022, doi:10.3390/md20030199_

Round 1
Reviewer 1 Report
This review presents numerous flaws and poor grammar in certain sentences. The absence of line numbering difficult the review comments.
The references used were not adequate in several cases and presented numerous flaws in the reference list (species not in italics (30, 53, 56, etc), first name of species in minuscule (34), name of journal missing, despite DOI provided (12, 13, 18, 21…., 54, 63, …. Etc, etc).
The subject of ‘Emerging Biotoxins’ is not new, but strangely, azaspiracids, toxins that are common in northern Europe since the 1990s were considered here as emerging toxins. This classification attains the ridicule of azaspiracids above the regulatory limit in Sweden and not causing any human intoxications (‘0’ poisonings) to appear in a table with reported intoxications over the last 15 years.
Gastroenterites after shellfish consumption were common in Europe for decades, but only in the 1980s some of these were attributed to DSP and in the 1990s to AZAs. This is not related to their ‘emergence’ in northern Europe (where contamination with AZA is endemic, but to the lack of adequate scientific knowledge and analytical methods before that time. The ‘emerging biotoxins’ are mainly related to the tropicalization of the Mediterranean and Macaronesia regions (southern Europe only).
Introduction:
«Every year, there are almost 2,000 cases of human intoxication upon shellfish or fish consumption with a mortality rate of 15% [3].» This is an outdated statement related solely to PSP intoxications, which present indeed a high mortality. Overall, marine toxins rarely are lethal. Also, this statement is not in accordance with the «estimated 10,000 to 50,000 intoxications per year worldwide (page 7)» just for ciguatera. And also, reference [3] is about «toxins Pinnatoxin-G and High Levels of Esterified OA Group», clearly unrelated to statistical data about human intoxications. This sentence is an example of how the authors were not very careful in writing the manuscript and providing adequate citations.
The « maximum levels in seafood [5,6].» are not up to date (legislation from 2004 and 2013 only), as pectenotoxins were already removed from the EU legislation in 2021. Also, the following sentences about the lipophilic groups are not correct after pectenotoxins were removed from EU legislation.
« The STX group is composed by four subgroups of 57 analogues:» This is not in agreement with the four groups described next, where 14 compounds were detailed. These are indeed the most common analogues in bivalves, but one relevant in Europe is missing: GTX6. The remaining 57 analogues are unrelated to this so-called ‘four groups’, and some exist only in freshwater habitats.
«Until 2013, all official toxicity determinations in EU countries were carried out through mouse bioassay MBA),…». This is far from the truth!!! ASP is monitored in Europe long before 2013, and never the mouse bioassay was used as a testing method (only HPLC). See for example COMMISSION REGULATION (EC) No 2074/2005. Also, some countries replaced mouse bioassays for DSP/PSP before 2013.
«AZAs group which nowadays consists of more than 60 compounds with different toxicity [14].» This article [14], identifies Azadinium species in the Pacific northwest and the authors claimed the research ‘demonstrates the need to assess their toxicity….’. This article related to AZAs only presented species, but not their toxins. The use of this reference was not appropriate for the introduction, nor for the next sub-sections, such as section 2.2.
From the comments so far, I believe this review does not constitute a reliable piece of research and should be rejected.
Author Response
Reviewer 1
This review presents numerous flaws and poor grammar in certain sentences. The absence of line numbering difficult the review comments.
The manuscript was substantially rewritten considering all the reviewer comments. Line numbering is now included.
The references used were not adequate in several cases and presented numerous flaws in the reference list (species not in italics (30, 53, 56, etc), first name of species in minuscule (34), name of journal missing, despite DOI provided (12, 13, 18, 21…., 54, 63, …. Etc, etc).
The referee is right. This was a bibliography program error when we formate the manuscript. We have carefully checked now. We hope to meet the reviewers’ expectations.
The subject of ‘Emerging Biotoxins’ is not new, but strangely, azaspiracids, toxins that are common in northern Europe since the 1990s were considered here as emerging toxins. This classification attains the ridicule of azaspiracids above the regulatory limit in Sweden and not causing any human intoxications (‘0’ poisonings) to appear in a table with reported intoxications over the last 15 years.
In table 1 we included a few cases of azaspiracid poisoning in the last 15 years.
In this paper when we refer to emerging azaspiracids (AZAs), we are considering the non-regulated ones (excluding AZA1-3 which have already been regulated). To avoid misunderstandings, we have re-written all sentences concerning this issue and highlighted this approach through the manuscript.
Gastroenterites after shellfish consumption were common in Europe for decades, but only in the 1980s some of these were attributed to DSP and in the 1990s to AZAs. This is not related to their ‘emergence’ in northern Europe (where contamination with AZA is endemic, but to the lack of adequate scientific knowledge and analytical methods before that time. The ‘emerging biotoxins’ are mainly related to the tropicalization of the Mediterranean and Macaronesia regions (southern Europe only).
We agree with the reviewer that AZAs is not currently a problem in the EU. The toxigenic blooms of Azadinium (or Am. languida) are rather rare events and/or blooms are not persistent enough to cause significant shellfish toxin levels. We know that the intoxication risk is low. However, emerging AZAs can be in molluscs and this paper we wanted to value it. We did not mention that emerging AZAs constitute a health risk. We just wanted to highlight that besides AZA1-3, other non-regulated AZAs (and therefore emerging AZAs) can coexist in molluscs, although in small amounts. For example: AZA-4, AZA-5, AZA-6 AZA-11 and AZA-43 [1–3]. We have not concluded that emerging AZAs constitute a risk, we have just included the AZAs section to assess their risk. We apologise if this section was not clear enough. We hope this is clear in the new version of the manuscript.
Introduction:
«Every year, there are almost 2,000 cases of human intoxication upon shellfish or fish consumption with a mortality rate of 15% [3].» This is an outdated statement related solely to PSP intoxications, which present indeed a high mortality. Overall, marine toxins rarely are lethal. Also, this statement is not in accordance with the «estimated 10,000 to 50,000 intoxications per year worldwide (page 7)» just for ciguatera. And also, reference [3] is about «toxins Pinnatoxin-G and High Levels of Esterified OA Group», clearly unrelated to statistical data about human intoxications. This sentence is an example of how the authors were not very careful in writing the manuscript and providing adequate citations.
Thank you for the comment, we appreciate it. We have updated the sentence as follows:
“The main entrance route to humans of these toxic substances is through consumption of contaminated seafood and thus, a high number of intoxications occur every year”.
The « maximum levels in seafood [5,6].» are not up to date (legislation from 2004 and 2013 only), as pectenotoxins were already removed from the EU legislation in 2021. Also, the following sentences about the lipophilic groups are not correct after pectenotoxins were removed from EU legislation.
Thank you for the information. We have updated this part of the manuscript including the new legislation for pectenotoxins. In addition, the remaining part of the section and figure 1 was updated.
« The STX group is composed by four subgroups of 57 analogues:» This is not in agreement with the four groups described next, where 14 compounds were detailed. These are indeed the most common analogues in bivalves, but one relevant in Europe is missing: GTX6. The remaining 57 analogues are unrelated to this so-called ‘four groups’, and some exist only in freshwater habitats.
We thank the reviewers' comment, the sentence has been corrected and updated: “The legislated group of hydrophilic toxins is comprised of two distinct groups: saxitoxin (STX) and domoic acid (DA) and their derivatives. The STX group is composed of four subgroups: the C group, N-sulfocarbamoyl-gonyautoxins 1–4, (C1, C2, C3 and C4); decarbamoyl gonyautoxins 1–4 (dcGTX1, dcGTX2, dcGTX3 and dcGTX4); GTXs group, gonyautoxins 1–5 (GTX1, GTX2, GTX3, GTX4, B1 and B6) and the STX group (STX, decarbamoyl saxitoxin (dcSTX) and Neosaxitoxin (NEO))” – Page 2, lines 56-60.
«Until 2013, all official toxicity determinations in EU countries were carried out through mouse bioassay MBA),…». This is far from the truth!!! ASP is monitored in Europe long before 2013, and never the mouse bioassay was used as a testing method (only HPLC). See for example COMMISSION REGULATION (EC) No 2074/2005. Also, some countries replaced mouse bioassays for DSP/PSP before 2013.
Sorry for the mistake, we meant “ all official toxicity determinations in EU countries for lipophilic toxins were carried out through MBA”. The sentence was corrected.
«AZAs group which nowadays consists of more than 60 compounds with different toxicity [14].» This article [14], identifies Azadinium species in the Pacific northwest and the authors claimed the research ‘demonstrates the need to assess their toxicity….’. This article related to AZAs only presented species, but not their toxins. The use of this reference was not appropriate for the introduction, nor for the next sub-sections, such as section 2.2.
The reviewer is right. We have deleted this reference from the manuscript.
From the comments so far, I believe this review does not constitute a reliable piece of research and should be rejected.
The manuscript was improved. I hope the new version of the manuscript is considered for publication in Marine Drugs.
Reviewer 2 Report
The manuscript of Otero and Silva reviews the main detection methods for emerging algal toxins of the European area. After introducing the main class of toxins, their distribution and few knowledge on their toxicity, the authors describes the main methods. Even though this topic is already covered by other reviews, detection of harmful algal toxin is an important aspect. However, several issues should be considered in the manuscript, before being acceptable for its publication in Marine Drugs.
A general comment is that in the review the authors did not clearly express that the mouse bioassay has been substituted by chemical analytical analysis (namely HPLC), that now is the reference method. On the contrary, in some chapter authors state that no reference methods are currently available.
Another general comment is that chemical methods have only been inserted in table 3, but not sufficiently discussed in the text. This part should be implemented.
A third general comment is that literature citation is sometime incomplete or misleading. For instance, in the azaspiracid part, the in vivo studies were wrongly cited, omitting some of them. Similarly, some different in vitro studies have not been included in the citations. In the palytoxin section, citations 80-81-82 are not pertinent (other different more pertinent articles/reviews should be cited here) and authors should be aware that O. siamensis is not a producer of palytoxin, but of ostreocins.
I also suggest the authors to add the law references in the introduction and figure 1 for all the legislated toxins.
Author Response
Reviewer 2
The manuscript of Otero and Silva reviews the main detection methods for emerging algal toxins of the European area. After introducing the main class of toxins, their distribution and few knowledge on their toxicity, the authors describes the main methods. Even though this topic is already covered by other reviews, detection of harmful algal toxin is an important aspect. However, several issues should be considered in the manuscript, before being acceptable for its publication in Marine Drugs.
Thank you for your comment and the time investing in reviewing the manuscript.
A general comment is that in the review the authors did not clearly express that the mouse bioassay has been substituted by chemical analytical analysis (namely HPLC), that now is the reference method. On the contrary, in some chapter authors state that no reference methods are currently available.
We thank the reviewers' comment, we were referring to all legislated toxin groups since emerging toxins an interlaboratory validation is still pending for a reference method determination. For the sake of clarity, we altered the sentence: “Until 2013, for legislated toxin group of lipophilic toxins, all official toxicity determinations in EU countries were carried out through mouse bioassay (MBA), but due to ethical and technical concerns, progressively fell into disuse and toxin determinations were replaced by chromatographic hyphenated techniques, supported by the update of the legislation” – Page 2 lines 63-68.
Another general comment is that chemical methods have only been inserted in table 3, but not sufficiently discussed in the text. This part should be implemented.
We thank the reviewers’ input to improve the manuscript.
A third general comment is that literature citation is sometime incomplete or misleading. For instance, in the azaspiracid part, the in vivo studies were wrongly cited, omitting some of them. Similarly, some different in vitro studies have not been included in the citations.
Thank you for the comment. We have corrected and included a more available bibliography concerning in vivo and in vitro studies for AZAs. However, we are willing to include some specific publications else if the reviewer prefers. The new paragraphs are shown below:
“The mode of action of AZAs in humans is unknown, although in vivo studies in mice showed that AZA1 is absorbed and distributed, being detected in spleen, kidney, lung, heart and liver, and brain [7,8]. AZA1 has been found to have cardiotoxic potential in rats. After repeated i.p. administration of sublethal doses, the rats displayed signs of heart failure and alteration of myocardium structure [9]. Oral toxicity of AZA1 towards mice indicates that single oral doses causing lethality vary from 250 to 600 μg/kg [10–12]. To date, in vitro potencies are reported as AZA-2 > AZA-6 > AZA-34 ≈ 37-epi-AZA-1 > AZA-8 ≈ AZA-3 > AZA-1 > AZA-4 ≈ AZA-9 > AZA-5 ≈ AZA-10 > AZA-33 > AZA-26 [13]”.
“Regarding detection methods, research has been focused on the production of monoclonal and polyclonal antibodies against AZA [14,15]. These have been developed into a competitive enzyme-linked immunosorbent assay (ELISA) [16] a magnetic bead/electrochemical immunoassay [17], and an immunosensor [18] for the polyclonal antibodies. The monoclonal antibodies were also used by the different research laboratories to develop an immunoassay for the detection of these biotoxins such as a microsphere/flow fluorimetry-based immunoassay [19,20]”.
In the palytoxin section, citations 80-81-82 are not pertinent (other different more pertinent articles/reviews should be cited here) and authors should be aware that O. siamensis is not a producer of palytoxin, but of ostreocins.
Thank you for the comment. Citations were updated as follows: “Despite this, due to its high toxicity, human fatalities have been well documented [21–25]”. And O siamensis is not considered a palytoxin producer in the new version of the manuscript.
I also suggest the authors add the law references in the introduction and figure 1 for all the legislated toxins.
Thank you for the recommendation. The legislation was included.
Reviewer 3 Report
This review focuses on EU emerging toxins that have not drawn enough attention and are essential. It is a good and necessary review, but in order to be published, there are some things to be corrected.
Because of the lack of line numbering, I can't write (as I commonly do) the line number and the observations. I believe this is an editor's error. Nevertheless, I am sending the attached document with the highlighted statements, mostly in yellow, and with comments (in most of them. Not when I decided it was too obvious).
I noticed a worrying lack of attention in writing. It is carelessness that should be avoided. And this is noticeable from the Abstract. It is a pity that such a thorough source investigation loses worth due to this lack of attention, like writing "dinnoglagelate" instead of dinoflagellate.
In general: English must be reviewed, preferably by an expert or a native speaker. But also the use of punctuation (which is basically the same in other languages, with few exceptions) should be thoroughly checked. There are dozens of commas sitting in unexpected places and lacking in others, and the same applies to most of the punctuation. There are also periods in bizarre places. And also very weird capitalization.
The authors use many word contractions that should be avoided in written formal texts. Also, some words were changed (like "them" instead of "then").
Figures: the shadow letters are difficult to read. The figure was prepared in MS Word with the formatting marks button on and then copied as an image. Every single space has a dot on it and makes it very hard to read.
I recommend placing a figure with the toxin structure in each chapter and not all at the end. Or, at least, mention this figure in the text.
Fig. 2: Please find better resolution images for A, C, and F.
Tables: these are very hard to read. The double spacing and the justified text, plus the weird use of periods and the non-clear way of presenting the information, are not helpful. A table should clearly present the information that the reader can get almost at a glance. Please re-design your tables so that they are clear.
Introduction: 2,000 cases of human intoxication, but the reference is not correct. And also, this information is old. Why do the authors are not mentioning dcGTX1/4? And GTX5 should be B1, and what about B2 (GTX6)? Are these not regulated? Because, according to recent literature, they are potent toxins as well (primarily dcGTX1/4).
It is vital to highlight the importance of these "toxins" and why they are considered toxins even if no human intoxication has ever been reported. Bring this to the discussion.
Conclusions: there is a lot to be concluded about these emerging toxins. The review could open a very healthy discussion on the following steps to be made by the EU agencies. I am sure that a better conclusion will create a more valuable review.
Author Response
Review 3
This review focuses on EU emerging toxins that have not drawn enough attention and are essential. It is a good and necessary review, but in order to be published, there are some things to be corrected.
Because of the lack of line numbering, I can't write (as I commonly do) the line number and the observations. I believe this is an editor's error. Nevertheless, I am sending the attached document with the highlighted statements, mostly in yellow, and with comments (in most of them. Not when I decided it was too obvious).
We thank the reviewers' comments. However, the pdf file was not provided by the journal. Also, we added line numbering to facilitate the review.
I noticed a worrying lack of attention in writing. It is carelessness that should be avoided. And this is noticeable from the Abstract. It is a pity that such a thorough source investigation loses worth due to this lack of attention, like writing "dinnoglagelate" instead of dinoflagellate.
Our apologies and thank you for the comment. All manuscript was improved.
In general: English must be reviewed, preferably by an expert or a native speaker. But also the use of punctuation (which is basically the same in other languages, with few exceptions) should be thoroughly checked. There are dozens of commas sitting in unexpected places and lacking in others, and the same applies to most of the punctuation. There are also periods in bizarre places. And also very weird capitalization.
The authors use many word contractions that should be avoided in written formal texts. Also, some words were changed (like "them" instead of "then").
We thank the reviewers' comments, the manuscript was proofread.
Figures: the shadow letters are difficult to read. The figure was prepared in MS Word with the formatting marks button on and then copied as an image. Every single space has a dot on it and makes it very hard to read.
Thank you for the comment. Figures were improved.
I recommend placing a figure with the toxin structure in each chapter and not all at the end. Or, at least, mention this figure in the text.
We thank the reviewers' recommendation, Figures are mentioned along with the text.
Fig. 2: Please find better resolution images for A, C, and F.
Images have been improved.
Tables: these are very hard to read. The double spacing and the justified text, plus the weird use of periods and the non-clear way of presenting the information, are not helpful. A table should clearly present the information that the reader can get almost at a glance. Please re-design your tables so that they are clear.
Thank you for the comment. The table information is structured in columns with the information to describe in the upper row. Abbreviations and legends are included. Double spacing was reduced.
Introduction: 2,000 cases of human intoxication, but the reference is not correct. And also, this information is old. Why do the authors are not mentioning dcGTX1/4? And GTX5 should be B1, and what about B2 (GTX6)? Are these not regulated? Because, according to recent literature, they are potent toxins as well (primarily dcGTX1/4).
We thank the reviewers’ comment, sentence corrected and updated: "The main entrance route to humans of these toxic substances is through consumption of contaminated seafood and thus, a high number of intoxications occur every year [5,6]".
It is vital to highlight the importance of these "toxins" and why they are considered toxins even if no human intoxication has ever been reported. Bring this to the discussion.
We agree with the reviewer, the introduction was improved:
Page 1 lines 29 -37: “Marine biotoxins are natural toxic metabolites usually produced during harmful algal blooms (HABs) phytoplankton that get accumulated in marine organisms and migrate along the food chain [1]. A HAB is characterized by a rapid proliferation of phytoplankton, the so-called red tides. Still, their forming circumstances are to be disclosed, although the rise of water temperature and anthropological intervention are pointed as main triggers [2]. These secondary compounds are produced to give their producers a competitive advantage against similar species and, also provide a defence against predators [3]. Their deleterious effects can impact an entire ecosystem leading to high fish mortality, affecting the fishery, the aquaculture and threatening public health [4].“.
Page 2 lines 87-91: “Even though human poisoning events were not associated in some groups, their toxicity has already been proven in animal models or cell assays. Also, effects of a chronicle exposure are scarce or absent, with unknown consequences The scarcity of reports could be also given their novelty in European territory, health professionals are not aware of this new threat.”
New references added:
Cembella, A. Chemical ecology of eukaryotic microalgae in marine ecosystems. Phycologia 2003, 42, 420–447, doi: 10.2216/i0031-8884-42-4-420.1.
Griffitth, A.W., Gobler, C.J. Harmful algal blooms: A climate change co-stressor in marine and freshwater ecosystems. Harmful Algae 2020, 91, 101590. Doi: 10.1016/j.hal.2019.03.008.
Townhill, B. L., Tinker, J., Jones, M., Pitois, S., Creach, V., Simpson, S. D., Dye, S., Bear, E., and Pinnegar, J. K. Harmful algal blooms and climate change: exploring future distribution changes. – ICES Journal of Marine Science, 75: 1882–1893
Conclusions: there is a lot to be concluded about these emerging toxins. The review could open a very healthy discussion on the following steps to be made by the EU agencies. I am sure that a better conclusion will create a more valuable review.
We thank the reviewers' suggestion to improve the manuscript, conclusions have been improved, please see page 18 lines 637-640: “Although the number of poisoning cases reported worldwide is considerable, there is a lack of epidemiological studies and the effects of chronic exposure to these compounds are not yet known. Sensitizing the health community is crucial, as is the establishment of an international network for predicting and signalling blooms and poisoning cases.”
Reviewer 4 Report
The article " Emerging Marine Biotoxins in European Waters: Potential Risks and Analytical Challenges" by Otero and Silva reviews the status of emerging phycotoxins in European seas and oceans, and the development of analytical methods to monitor their levels in different matrixes.
In general, the review is well constructed, very well researched and fairly well illustrated. I found a few typos, but the fact that the draft does not have numbered lines does not facilitate corrections.
The 3 tables are very complete, but their length makes them a bit difficult to read.
Author Response
Reviewer 4
The article " Emerging Marine Biotoxins in European Waters: Potential Risks and Analytical Challenges" by Otero and Silva reviews the status of emerging phycotoxins in European seas and oceans, and the development of analytical methods to monitor their levels in different matrixes.
In general, the review is well constructed, very well researched and fairly well illustrated. I found a few typos, but the fact that the draft does not have numbered lines does not facilitate corrections.
We thank the reviewers' comments, we added line numbering for review facilitation. English has been corrected.
The 3 tables are very complete, but their length makes them a bit difficult to read.
Thank you for the comment. Table information is structured in columns with the information to describe in the upper row. Abbreviations and legends are included. Double spacing was reduced.
Round 2
Reviewer 1 Report
I am pleased with the improvments made to the mnuscript.
Reviewer 2 Report
I appreciate that my previous suggestions/comments have been considered by authors and, indeed, the manuscript has been improved.
However, literature citations still need to be implemented. Just few examples:
Azaspiracids: the only comparative study of the oral toxicity of AZA-1, -2 and -3 should be cited (Pelin et al. 2018 Toxicol lett). In addition, since the study of Aune is related to a combination with okadaic acid, this aspect should be explained in the text and, therefore, authors should cite also an in vivo study carried out combining AZA and yessotoxin (Aasen et al., 2011 Toxicon).
Palytoxin: at lines 262 and 263 I would include also the following references (Durando et al. 2007 Euro Surveill; Del avero et al., 2012 Ann Ist Super Sanita; Tubaro et al 2011., Toxicon 478-95).